# Community Perceptions of Arsenic Contaminated Drinking Water and Preferences for Risk Communication in California’s San Joaquin Valley

**DOI:** 10.3390/ijerph20010813

**Published:** 2023-01-01

**Authors:** Hollynd Boyden, Mayela Gillan, Javier Molina, Ashok Gadgil, Winston Tseng

**Affiliations:** 1Health Research for Action, School of Public Health, University of California, Berkeley, CA 94720, USA; 2Department of Civil and Environmental Engineering, University of California, Berkeley, CA 94720, USA

**Keywords:** risk communication, community education, community engagement, rural community, drinking water, arsenic

## Abstract

Due to chronic exposure to elevated levels of arsenic in drinking water, thousands of Californians have increased risk for internal cancers and other adverse health effects. The mortality risk of cancer is 1 in 400 people exposed to above 10 μg/L of arsenic in their drinking water. The purpose of this community assessment was to understand the perceptions and awareness of the residents and public water representatives in rural, unincorporated farming communities of color in San Joaquin Valley, California. In our research, we asked 27 community informants about their (1) available water sources, (2) knowledge about the health impacts of arsenic, and (3) preferences for risk communication and education regarding the health impacts of arsenic-contaminated drinking water. Through our qualitative coding and analysis, we found that most community informants indicated that there was limited community awareness about the health effects of drinking water with elevated arsenic levels. Preferences for risk communication included using in-language, culturally relevant, and health literate health promotion strategies and teaching these topics through the local K-8 schools’ science curriculum with a language brokerage approach to transfer student knowledge to family members. Key recommendations include implementing these communication preferences to increase community-wide knowledge about safe drinking water.

## 1. Introduction

Inorganic arsenic, a form of the elements typically found in soil, bedrock, and associated groundwater is the most toxic form of arsenic to humans [1]. An estimated 220 million people worldwide are exposed to drinking water with arsenic above the World Health Organization’s provisional guideline and the U.S. EPA Maximum Contaminant Level (MCL), 10 micrograms per Liter (μg/L) [1,2,3]. High levels of arsenic in drinking water have been found in countries such as Argentina, Bangladesh, Chile, China, India, Mexico, and the United States [1]. Within the United States, high levels of naturally occurring arsenic have also been found in the groundwater aquifers used for drinking water supplies, particularly in the states of Utah, Nevada, Arizona, and California [4]. Health effects attributed to chronic exposure to arsenic, most commonly through ingestion of contaminated water, have been studied extensively and arsenic exposure has been associated with lung, bladder, skin and kidney, cancers along with skin lesions and impairments to the cardiovascular and nervous systems [4,5,6,7,8,9,10,11]. The International Agency for Research on Cancer has labeled arsenic as a Group 1 known human carcinogen [12], and according to a recent analysis and review, the excess mortality risk due to cancer for exposure above 10 μg/L is 1 in 400 people with an increase in mortality risk for exposure above 50 μg/L at 1 in 81 people [5]. 

From 2014–2015, it was estimated that 55,000 people across California were exposed to arsenic contamination above the legal level (10 μg/L) through their local community water systems [13]. Community water systems and domestic/private wells are the primary sources of drinking water for residents of California A community water system is defined as a public water system that serves the same population year-round and must have at least 15 service connections or serve an average of at least 25 people for at least 60 days a year [14]. Community water systems can be further categorized by the number of people they serve. A small water system serves 501–3300 residents and a state small water system serves 5–14 service connections and does not serve an average of at least 25 people for at least 60 days a year [15]. Community water systems in California require domestic water supply permits which are regulated by the system operators and must maintain compliance with state and federal MCLs to ensure safe drinking water. For rural communities, small community water systems are overseen by the California State Water Resources Control Board, the Department of Environmental Health, or the California Public Utilities Commission and the state small water systems are overseen by the Department of Environmental Health at the county level [16]. For these small water systems, each county sets its own requirements for water quality testing. Domestic/private wells provide water for residential use and serve less than 4 connections and are not regulated by the California State Water Board Division of Drinking water, which makes it harder to track how many people could be at risk of drinking arsenic-contaminated drinking water. Local health officials and state regulators are concerned about ensuring safe drinking water for both community water systems and domestic/private well owners [17]. 

The nature of improving and regulating water quality in small community water systems or state small water systems in rural areas in the US and around the world proves to be a worthy challenge. The United Nations’ Sustainable Development Goal 6 “Ensure access to water and sanitation for all” emphasizes the importance of investing in safe drinking water for community benefits including the town’s continued growth and the residents’ continued empowerment [18]. However, limited funding for maintaining water treatment systems and water distribution systems, remain primary challenges for small community water systems to achieve this goal. Since rural communities are often spread across many miles, it is difficult and expensive to maintain piped water to each service connection and to guarantee the delivery of high-quality water. For unincorporated communities that rely on county-level funding, it is even harder to provide infrastructure updates or routine maintenance [19]. A lack of funding often poses a challenge in meeting regulatory standards for water quality [16]. Violations of the MCL can result in increased monitoring of the system, distribution of notices to the public, and sometimes fines imposed on the services district which can take away from already minimal financial resources [20,21,22,23,24].

Our research focused on two communities in the San Joaquin Valley (SJV) of California. The SJV is located in central California with the Coast Range to the west and the Sierra Nevada Mountains to the east. This region is of particular concern because it has two of the most contaminated aquifers in the United States [25]. Due to the reliance on groundwater aquifers as the source of drinking water, the residents of the SJV are at high risk of ingesting water contaminated with arsenic above its MCL, unless the drinking water is properly treated [26]. studies have shown that the highest cumulative contaminant concentrations in community water systems in California, including contamination from arsenic, occur there [27]. In particular, it is estimated that the SJV is the region with the most people exposed to arsenic across California [28].

Furthermore, the SJV is home to a diverse population of residents with the majority of residents living in largely low-income, rural communities supporting much of California’s agricultural land [29]. The SJV communities also have some of the highest rates of poverty and minority populations in the state of California [25]. According to the 2021 American Community Survey conducted by the U.S. Census Bureau, 51.4% of the population identifies as Hispanic/Latino, compared to 31.1% identifying as White alone within the eight counties (Fresno, Madera, Merced, Kern, Kings, Stanislaus, San Joaquin, and Tulare) in the San Joaquin Valley [30]. This region is also home to nine Indigenous tribes who live throughout the area, often in rural areas or unincorporated towns. Communities of color (Hispanic/Latinos and Non-Hispanic people of color compared to Non-Hispanic White) are often served by community water systems, state small water systems, small water systems, and domestic/private wells are associated with greater drinking water contamination [31]. One report estimates that among the residents who are served by community water systems that have received “out of compliance” violations, 57% identify as Hispanic/Latino while only 35.8% identify as White [16]. Additionally, studies in the SJV have shown that community water systems that serve low socio-economic status communities have higher arsenic levels and higher odds of receiving a maximum contaminant level violation [5]. Together these studies indicate the environmental injustice that exists in the SJV where low-income, Hispanic/Latino people, tribal nation members, and/or Black Americans living in rural areas are disproportionately exposed to higher levels of arsenic through their drinking water sources. These environmental health disparities in the SJV were some of the drivers of our current project where we used a health equity lens to assess the community perceptions of arsenic-contaminated drinking water and preferences for risk communication about the possible health effects.

Our study represented a major community assessment component of a larger study aimed at designing an effective and acceptable arsenic remediation technology. An arsenic remediation technology of this nature would reduce current injustices regarding the availability of clean drinking water across the world. Literature has shown that solutions to arsenic contamination must adapt to the social, economic, cultural, and institutional contexts of the community it serves [32]. There are currently two major approaches to reducing arsenic exposure at the community scale: (1) the use of an alternative safer water source, and (2) reducing the arsenic concentration in the current source [32]. Across the globe, we have seen communities use alternate sources by switching to deeper wells instead of shallow wells, in hopes of avoiding arsenic-contaminated shallow aquifers [33] or by purchasing arsenic-free water from “water vendors” [34]. To reduce the arsenic concentration in a contaminated source, a few different remediation technologies have been proven to be effective including reverse osmosis, oxidation followed by adsorption and filtration, and electrolytic methods, among many others [35,36]. There are also pilot studies for different remediation technologies that would be operated at the local community water system [35,37,38]. 

One such technology is Electrochemical Arsenic Remediation (ECAR) invented in 2010 by Gadgil and his colleagues at the Lawrence Berkeley National Laboratory, operated by the University of California [32,38]. The ECAR technology was successfully implemented to serve 5000 people daily in a rural community of Dhapdhapi in West Bengal, India in 2016, and lowered the arsenic levels from 250 µg/L to below 3 µg/L [38,39]. A second community-scale installation of ECAR, this one fully powered with rooftop solar, started operation in the small town of Bahraich in North India in early 2021. The Gadgil Lab at UC Berkeley concurrently worked on a significantly improved version of the ECAR technology, called Air Cathode Assisted Iron Electrocoagulation (ACAIE) which is hundreds of times faster (and therefore more compact and less cumbersome) than the original ECAR technology. ACAIE, developed in 2019, is also more efficient and effective at removing arsenic from contaminated water [40]. ACAIE aims to treat water at less than $0.25 per liter to provide more affordable access to safe drinking water. After the successes in India, the Gadgil Lab at UC Berkeley turned its interest to solving the arsenic crisis in its home state: California. Currently, this ACAIE technology is in the field-testing phase, which began in 2022 in a community located within the SJV. Our study represented a major community assessment component of the overall ACAIE remediation technology’s ongoing community-engaged field testing in SJV.

Community-engaged research prioritizing participatory approaches and community perspectives has been identified in environmental health literature across the world to be critical to identifying and developing sustainable community-relevant strategies to address clean water and other environmental health issues [41,42,43,44,45]. Prior studies have shown that engaging community leaders and residents in the design and implementation of environmental health interventions has facilitated the success of these interventions. Interventions and technologies that are developed in partnership with communities and incorporate community perspectives in their design have been shown to be more readily accepted, and have longer-lasting impacts on community health. Therefore, this research centered a community-based participatory research (CBPR) approach to support three long-term goals of the larger transdisciplinary project: (1) to better develop a community education and outreach program that increases community knowledge about the dangers of arsenic-contaminated drinking water, (2) to better facilitate the acceptance of the ECAR-ACAIE arsenic mitigation strategy by trying to learn the community’s relevant concerns so we might address them during future implementations, and (3) to inspire behavior change regarding water consumption by community members. This work was conducted in two rural, unincorporated communities within the SJV, California. This project was built on pre-existing partnerships with two rural SJV communities: the unincorporated, agricultural regions of Lemoore and Allensworth, both with a high percentage of residents of color including Hispanic/Latino people, tribal nation members, and/or Black Americans. 

Evidence-based CBPR approaches include iterative processes through building relationships with community partners and promoting co-learning and information sharing, among other practices [43,44]. In this study, we developed local partnerships and generated information-sharing strategies in order to better develop a risk communication and education program that could be implemented to increase knowledge and awareness of a topic of concern: health risks associated with arsenic-contaminated drinking water. The purpose of this community health assessment as part of the larger transdisciplinary project was to understand (1) the awareness and perceptions of at-risk rural communities about their water sources and the health impacts of arsenic contamination, and (2) the preferences for risk communication and education regarding health impacts of arsenic-contaminated drinking water.

## 2. Materials and Methods

We used a participatory approach to conduct a qualitative community assessment, including 27 community informant (CI) interviews from September 2020–June 2022. Out of the 57 potential participants, 27 (47%) completed the telephone interview. Our Cis included diverse perspectives from community leaders and residents, school district members, and public water representatives from various organizations (see Table 1). This participatory, community assessment study explored community experiences and knowledge of safe drinking water and arsenic risks and identified barriers and preferences for community-wide risk communication and education strategies. The study results will be used to design and develop meaningful and relevant community-wide environmental health education initiatives to raise community awareness about the quality of their local water systems and water safety.

### 2.1. Study Area

As stated in the introduction above, the San Joaquin Valley of California has been found to have high levels of water contaminants—including arsenic—with community water systems and domestic/private wells that serve populations of color (Hispanic/Latino, tribal nation members, Black Americans, etc.) more frequently receiving maximum contaminant violations compared to their White counterparts. We partnered with Allensworth and the unincorporated region of Lemoore, both communities located in the SJV who are at high risk of drinking arsenic-contaminated water (see Figure 1). Both communities are unincorporated, agricultural rural towns with about 500 residents each with a majority population of residents of color. About 95% of Allensworth residents identified as Hispanic/Latino while the residents of the unincorporated region of Lemoore largely identified as tribal nation members or other minority populations. In these agricultural communities, people often live far apart from each other with limited or no wifi/technology access. Many have low literacy and/or limited English proficiency.

### 2.2. Sampling Plan and Recruitment

Participants were recruited using both purposive and snowball sampling strategies. These qualitative sampling strategies are particularly effective for exploratory/formative studies such as our community assessment where little is known about these communities and for reaching hard-to-count, small rural underserved populations that do not trust researchers [46]. We purposively sampled a variety of community informants as part of the qualitative triangulation of diverse community perspectives to confirm the community issues identified by the assessment could be confirmed by multiple informant sources. At the same time, this would collectively allow for the most comprehensive, in-depth understanding of the issues of water safety and water systems in these communities. These informants included diverse community stakeholders such as local community residents and leaders (Hispanic/Latino, tribal nation, and Black), school district members (officials, teachers, and parents), community water service managers, and environmental health officers. Community informants were eligible if they were 18 or older and resided or served the greater San Joaquin Valley including Allensworth and the unincorporated region of Lemoore. Participant outreach occurred via phone, text, email, or in-person by the interview team or local community liaison/partners. We recruited a variety of community informants that lived or worked in these rural communities (community residents, community leaders, school district members, water service providers/experts) as part of the triangulation of community perspectives to ensure the community issues identified would be confirmed by multiple sources. Community informant recruitment ceased when thematic data saturation was observed in the concurrent data analyses. Triangulation and saturation of the sample strengthened the credibility of our study.

### 2.3. Data Collection

All the interviewers on the research team were trained to conduct interviews to be able to follow the same interview protocol in an accurate, consistent way. Community informants participated in a 30- to 60 min semi-structured open-ended telephone interview conducted by a member of the research team. Electronic or verbal informed consent was given to the interviewees via email or phone prior to proceeding with scheduling an interview. Before the beginning of each interview, the consent information was repeated verbally, and they were asked to reconfirm their consent. In the telephone interview which included 11 questions, participants were asked to elaborate on (1) their opinions and experiences with safe drinking water, (2) knowledge about arsenic risks in their communities, and (3) barriers and preferences for community-wide communication and education strategies. We asked two additional questions pertaining to the impact of the COVID-19 pandemic in 2020. All the interviews were recorded and transcribed. Interview notes were also taken during the interviews.

### 2.4. Data Analysis

Interview transcripts and notes were compiled for qualitative analysis. The constant comparative method was used as a technique for the qualitative thematic analysis and took into consideration key qualitative methodologies of triangulation, saturation, and credibility [47]. This method developed codes, examined relationships and interactions across descriptive and thematic codes, and compared the major themes that emerged from the coding categories. Qualitative analysis probed for parallel themes, particularly looking for knowledge and understanding of safe drinking water and associated arsenic risks, and barriers and preferences for risk communication and education to increase community awareness of arsenic-contaminated drinking water. The final codebook consisted of descriptive and thematic codes common across the 27 community informant (CI) interviews. Five research team members conducted the qualitative analysis and coding. These research team members were trained to use the same, consistent coding approach. Two to three research team members independently coded and then met together to discuss and compare each other’s coding for each of the transcripts from the CI interviews and come to an agreement with the descriptive and thematic codes for the data analysis. Inter-rater agreement for the interview transcripts was determined to ensure consistency in coding. For each interview transcript, if 80% agreement in coding consistency was not reached, the researchers discussed potential issues that arose and reached a consensus about these coding issues until consistency was reached. The coding of the transcripts was an iterative process. The codebook was revised as subsequent transcripts were coded and new codes emerged. Coding consistency was recalibrated as part of this iterative coding process. Code categories were connected and grouped through thematic coding, and the researchers identified major themes from the codes. In addition, we reached data saturation mid-way through the coding of the 27 informant interview transcripts but continued completing the analysis of all 27 transcripts. We analyzed substantially more than the number of interview transcripts needed to reach data saturation or when additional interviews did not provide any new data or information and started to repeat what was previously coded. The qualitative coding used the software Dedoose Windows version is 9.0.62.

## 3. Results

### 3.1. Summary of CI Characteristics

In total, 27 qualitative community informant interviews were conducted. Of the 27 community informants, 52% were female and 48% were male. 48% of participants reside or serve in the unincorporated region of Lemoore, California, 44.5% in Allensworth, California, and 7.5% from nearby areas in the San Joaquin Valley. The community informants have a range of backgrounds including community leaders, residents, and water professionals representing various organizations (Table 1).

### 3.2. Community Understanding of Arsenic Risk in Drinking Water

#### 3.2.1. Community Perceptions and Experiences with Arsenic Exposure in Drinking Water Systems

Community informants (CIs) expressed their awareness and perceptions of arsenic in their drinking water and shared their concerns about the failing water systems in their rural, farming communities of color. All CIs indicated they were aware of arsenic in their drinking water. Almost all participants also expressed serious concerns about the persistent risk to arsenic in their drinking water. Of those that did not express a negative perception, none of them expressed a positive perception either. One participant spoke of the ubiquitous nature of arsenic in their community stating: 

*“Arsenic is just prevalent in this area. It’s naturally occurring in our groundwater and so very likely to be present in public water systems, private wells, pretty much any groundwater source it’s likely to be found.”* (Public Water Representative)

Most CIs stated that the problem of arsenic contamination in their drinking water has been a long-term problem historically and continues today. Many CIs perceived that the water in their community was not safe for drinking and had concerns that the existing community water utility system was “archaic” and “unreliable.” One participant expressed their frustration saying, 

*“I mean, here in [town name], you never know what you’re drinking, because I mean, there’s so much that we don’t know.”* (Local School District Member)

Many CIs shared frustrations that community residents were being left to find their own solutions. Many CIs stated that a lot of people, including themselves, relied on bottled water for drinking and cooking due to the water contamination in the community water system or a local well, and expressed frustration at the injustice of this. Some CIs with the local schools or community organizations indicated they were able to implement temporary solutions such as point-of-use filters or free bottled water with support provided by organizations like Rural Community Assistance Corporation or Self-Help Enterprises. A few CIs discussed their hopes for a longer-term solution through digging and building a new community well that is arsenic free with funds from the state of California. However, they indicated they had been let down as no progress had been made by the state in the past ten years after funding for the community well was initially approved.

A few CIs from the local schools also expressed a concern that some of the school educational funds typically allocated for teachers and students were being reallocated in order to provide the students with bottled water. Less funds for education might affect the quality of education. However, regardless, all the CIs interviewed who were associated with the local schools were concerned about ensuring both safe drinking water and a good education for the students. One community informant stated,

*“It [Arsenic-contaminated water] affects the student’s learning capacity. I imagine what it does to someone that’s 40, 50, 60, 70 years old drinking the same water. Again, it deteriorates the [student’s] body from within and that has a big effect.”* (Tribal Nation Leader)

Funding a long-term, safe drinking water solution for their communities was a hope shared by many of the CI participants. However, financial constraints of these rural communities limited their options for clean water systems. Another challenge that a few community informants identified was that water quality changed depending on where a resident lived in the community. One participant acknowledged the challenges associated with arsenic contamination at a system level and expressed frustration at what arsenic contamination did to individuals: 

*“The mere fact that we have arsenic, any trace of arsenic in the water, is pushing a lot of very low-income people to go out and buy water [even though they cannot afford it].”* (Local School District Member)

#### 3.2.2. Lack of Understanding about Health Risks from Exposure to Arsenic and Other Water Contaminants in Drinking Water among Community Residents

Most CIs shared their lack of or misunderstanding of health concerns related to the arsenic contamination in drinking water within their rural, farming communities of color. A majority of CIs expressed that they were not knowledgeable of the severity and outcomes of the health effects caused by arsenic. Only some CIs correctly stated that arsenic contamination would affect their health negatively in the long term. Even though most CIs participants believed themselves to be far more knowledgeable than the rest of the residents in their communities about the impact of arsenic-contaminated water, most of them shared that the extent of their understanding of health concerns directly related to arsenic contamination was limited. These participants also mentioned that most of the residents in their communities were not aware of arsenic or other water contaminants in their drinking water and the associated health risks.

Only a few CIs specifically expressed concern that arsenic or poor water quality could be connected to cancer rates within their communities, which is a known health effect of long-term exposure to arsenic. A couple of CIs also expressed concern that some community residents did not understand the severity of long-term exposure to arsenic-contaminated water. While a few CI participants noted that their water has a poor taste and odor believing those qualities to be associated with arsenic, only one participant correctly indicated that arsenic is odorless and tasteless. A couple of CIs stated that some community residents liked the taste of the water and continued to drink it, even though they knew it was contaminated with arsenic and that it was not good for them. The CI participants indicated the community residents’ risk from arsenic exposure and its associated health effects were not perceived to be severe enough to deter them.

Limited knowledge of the health effects of arsenic contamination in drinking was paired with misinformation about other water contaminants. With respect to additional drinking water contamination sources, a few CI participants also identified lead, bacteria, nitrate, chromium, and pesticides as other possible contaminants that altered the states of water in their communities. One CI stated that they believed another contributor to limited knowledge about arsenic was that lead was discussed a lot more than arsenic in their communities. Although water contamination was a topic often discussed among the community informants, they reported that the residents of the wider community likely had little or no understanding about arsenic and other water contaminants in their community’s drinking water and the adverse effects on their health.

#### 3.2.3. Reasons for the Lack of Understanding about the Health Risks from Arsenic Contamination in the Drinking Water

According to multiple CIs, arsenic contamination to the community’s drinking water was often not a forefront concern for many in the community for a variety of reasons such as the lack of participation in regular community meetings about water quality, poverty, busy with their agricultural work, low literacy, and limited English proficiency. 

Some CIs cited one of the reasons as a lack of attendance at the monthly community action meetings where arsenic contamination in their drinking water was regularly discussed. One participant noted the importance of residents attending community action meetings where arsenic contamination was often discussed in order to stay informed about this issue by stating: 

*“There needs to be more awareness of the water and things in the community. They [the community] don’t get involved a lot, and then, they complain, and then, I tell them well, if you are going to complain, why don’t you go to the meeting?”* (Community Resident)

Behind this lack of attendance at community action meetings, some CIs indicated there were systemic barriers that must be considered including race, poverty, and language barriers. Some of the CIs believed that the poverty level, low literacy levels, limited English proficiency, and farm worker profession of the wider rural community puts these communities at a higher risk for misunderstanding the existing water safety crisis and therefore continued to consume arsenic-contaminated drinking water. Some CIs also believed that the residents of the wider community did not have adequate income to spend on purchasing bottled water and, at the same time, did not have the literacy level or English language proficiency to fully understand community announcements about contaminated water. Additionally, some CI participants indicated most community residents were just too busy trying to survive each day to support their families and be financially self-sufficient in the farming community. 

Additionally, CIs discussed the water board or water utility’s distribution of water district reports, statements, or flyers to the wider community. A few CIs stated that monthly water bills provided information on water contamination in English and Spanish, but stated that community residents might not take the time to read or understand these notices because of their busy work and family life in the farming community. Even though communications from the water board/utility were provided in both Spanish and English most of the time, many CIs mentioned they were worried that community residents did not read the reports due to low literacy levels in both English and Spanish. Another CI stated that they believed that a low understanding of safe drinking water was due in part to the minimal effort the public utility departments put into explaining the urgency of the issue to community residents in their reports. 

### 3.3. Barriers and Preferences for Risk Communication and Education

#### 3.3.1. Structural Barriers to Risk Communication and Education

In the interviews, CIs reported primary barriers to risk communication about environmental health risks associated with arsenic-contaminated drinking water in these rural, farming communities of color and their challenges with reaching community residents through existing communication and outreach approaches. These barriers included a lack of community residents attending or participating in the monthly community action meetings due to other work or family priorities and limited accessibility of the existing environmental health information materials related to arsenic and other contaminants in drinking water through existing public distribution channels. Additional barriers identified were difficulties in understanding existing high health literate informational materials due to language barriers and low literacy levels and the fragmented nature of these rural communities geographically and legally for outreach (i.e., incorporated regions, unincorporated regions, tribal nations).

Some CIs indicated regular community action meetings often discussed water quality and safety, but few or no community residents participated or prioritized going to them. This may have led to the lack of awareness of drinking water safety issues and was one of the primary barriers to risk communication. However, some CIs believed this lack of community participation in these meetings likely stemmed from deeper structural issues. Many CIs spoke of community residents having trouble attending meetings due to job schedules such as working long hours as farm workers or caring for their families. Succinctly put by one community informant: 

*“The challenge is always trying to figure out what’s the best sort of day and time to do a presentation. You have working families, you have people who work different times and days of the week and so it’s always trying to figure out when it’s going to work best for the majority of the community so that they are able to attend”* (Public Water Representative).

A number of CIs noted that even before the COVID-19 pandemic, attendance at community action meetings such as town halls was low and attendance worsened due to shelter-in-place during the pandemic. When meetings shifted to an online format during the COVID-19 pandemic, some CIs noted that limited internet access or limited ability to pay for internet-based access was an additional barrier that was linked to a decrease in community meeting attendance. One CI stated some of the challenges related to COVID-19 and meeting attendance:

*“We’ve tried a number of times to get people to dial in [for meetings], but that’s the other part of the problem. We don’t have good internet access here. It’s improving, but it’s not free. For somebody that has limited funds, I mean subscribing to an internet service is not viable. To try to get them to join the Zoom call, especially if they’re going to pay to join on their cell phone, makes no sense to them. We’ve had that as a challenge so far.”* (Community Resident)

CIs indicated that hopefully the COVID-19 pandemic will ease up and community meetings can return to in-person events which should lighten some of the barriers the community faces in attending these meetings. 

Other barriers that were cited by almost all CIs were (1) the lack of awareness about the severity of the issue from the public notifications about arsenic-contaminated water they received, (2) the language barriers in public communication and outreach, particularly to Spanish speakers, and (3) the high health literacy level of these public outreach materials distributed in these low literate, rural, farming communities of color. Some CIs shared that community organizations and water departments produce their own public notifications, mostly in the form of quarterly newsletters, with updates on water quality, although notification updates specific to arsenic and health were limited and only present when levels exceeded the safety standard. Some of the CIs stated that they had previously received communications from the water district about unsafe drinking water, but that these communication materials did not reflect an urgency about the arsenic contamination in their drinking water. A couple of CIs said that the water district also mailed and delivered bilingual materials to each household that notified them about their water quality. However, a couple of CIs also stated that they did not remember if these materials were in both English and Spanish or only in English. 

One CI stated that there was a “huge language gap” and that people with this Spanish language gap were the ones most adversely affected by the arsenic-contaminated water in their drinking water. Additionally, some of the participants stated that the community literacy level was low and that the high literacy level of the written materials presented in the community meetings or in public communications might not be the most effective way to disseminate this information. The dual language and literacy barriers were major communication and outreach issues. 

Another barrier unique to our study population was the fragmented nature of the towns geographically (i.e., incorporated regions, unincorporated regions, tribal nations) with people living far away from each other which is especially common in tribal nations. Additionally, due to tribal sovereignty, there are legal barriers that exist for tribal nations in these rural regions which complicate public water sharing because of the different regulations (state vs. federal) that govern water quality.

#### 3.3.2. Preferences for Education and Risk Communication about Drinking Water Safety

When asked about preferred strategies to improve education and communication about safe drinking water and water contaminants, CIs responded with several suggestions including the importance of educating the school children and having them bring this information back to their families and community residents. Additional suggestions were ensuring easy-to-understand bilingual materials are provided in English, Spanish, and indigenous languages, building trust with these communities for communication and outreach, and a few other public health education and promotion strategies. 

The most discussed strategy to improve risk communication about drinking water safety and health within these rural, farming communities of color was to provide educational materials about safe drinking water to the students at the local elementary and middle schools. Some CIs believed that by providing education about safe drinking water to the students they would then be able to take that drinking water safety and health information back to their parents and community residents, a practice known as language brokering. A CI that works directly with students affirmed that they found their elementary and middle school students often shared what they learned at school with their parents or guardians at home. One participant stated this philosophy simply:

*“Educate the young people, they’ll educate the community, and then we can move forward in a positive way that will benefit everybody.”* (Rural Community Leader)

Another CI acknowledged that this communication strategy was already being implemented through their community’s annual project-based learning summer workshops for local high school students and described its success by stating: 

*“The kids basically have a little seminar, and then [afterwards] they talk to their parents at least somewhat about it. Again, it depends on whether their parents are interested in or concerned about those kinds of issues or not and whether or not they’ll hear them out. From what I’ve noticed, it seems to be the case that the information usually at least gets to their parents.”* (Rural Community Leader)

One CI also suggested developing and holding annual water quality and environmental health training for elementary and middle school teachers, so they would be better prepared to discuss water topics and answer any questions that their students might have.

Many CIs explained the importance of having in-language educational materials that the students could bring back to their parents in English and Spanish since most of the students were bilingual and many parents were monolingual, Spanish speakers. By having bilingual materials, parents who were literate in Spanish and/or English could engage with their children’s school materials and for parents who were not literate in either language, these students could elect to read the materials to their parents in their preferred spoken languages. A few CIs cited this bilingual communication and outreach approach to have been successful thus far through their annual project-based learning summer workshops for high school students. They believed that it could work similarly if adapted to the school year curriculum and implemented in their local elementary and middle schools to improve community education about drinking water safety and health.

A majority of CIs suggested that relevant community educational materials developed by external community educators or partners about safe drinking water in English should also be translated in-language (e.g., Spanish) to have a larger impact on outreach to the broader community. A participant said that the community action meetings they have attended have been “only in English” which has led to a divide in the knowledge level between English speakers and non-English speakers. Many CIs stated that in-language communication and outreach should be applied to flyers or written materials that were sent out or distributed at these community action meetings or other local events so that these educational materials could reach as many people as possible across the community.

Many CIs described their towns as “close-knit communities” and ones that took care of one another. CIs also described their rural, farming communities of color as not trusting or receptive to outside assistance, even if the potential partner organizations were there to provide a solution for the arsenic crisis. A suggestion made by some CIs to external research partners was the importance of working closely with their community through cultivating relationships over time with trusted community residents who could advocate and support a research group’s presence in their communities.

A few other strategies to improve communication and outreach about arsenic-contaminated drinking water were presented by CIs. Some CIs agreed that a community action meeting with a visual aid or presentation would be the most effective educational format for meeting participants to increase their awareness about safe drinking water and water contaminants. Some participants also encouraged external community educators or partners to take part in their regular community or town events throughout the year to be able to reach and educate an even broader community. One participant stated: 

*“If you’re able to set up basically an event within the [community] event that’s already there, you’d be guaranteed a pretty good amount of community exposure, which then would allow you to hopefully get a good amount of community participation.”* (Rural Community Leader)

A few CIs indicated that these community action meetings should also include the incentive of free food. Previous town meetings that included free food were more widely attended by community residents compared to meetings that did not include free food. Another CI emphasized the importance of word-of-mouth as a primary communication in these rural, farming communities of color stating that,

*“Word of mouth could go a long way. If you get somebody, like if my Tia (aunt) or my comadre (godmother) would tell me, then I would trust her, right? Because I feel like she’s a trustworthy person. I think if you could just talk to as many people as you can and hopefully the word of mouth gets around.”* (Community Resident)

Finally, another CI said that using personal stories about arsenic in drinking water or other contaminated water from local town residents could go a long way because people related more to these personal stories from town residents.

## 4. Discussion

In this study, we learned from community informants in rural, farming communities of color about the little or no community awareness of health risks associated with arsenic-contaminated drinking water and the longstanding problems of their water systems in these unincorporated rural regions of San Joaquin Valley (SJV). We also learned about existing barriers to risk communication and education due to low literacy and limited English proficiency. Community informants made a number of suggestions for strategies to increase the risk communication and community education about the health risks of arsenic in drinking water including developing an innovative Science, Technology, Engineering, and Math (STEM) education curriculum about drinking water safety and health starting with educating local elementary school and middle school students in these communities. They mentioned that the knowledge these students learned in school or workshops could then be brought home and transferred to their families and community residents. Another suggestion was to incorporate linguistically inclusive risk communication messaging in Spanish and indigenous languages through community gatherings and trusted messengers. Additionally, some of the limitations of this study including sampling methods, interview techniques, and the impact of the COVID-19 pandemic are addressed.

### 4.1. Community Awareness and Barriers to Knowledge Acquisition

Our results identified the lack of awareness among rural community residents about health risks associated with arsenic, the predominant concerns of community residents, and the existing barriers to gaining knowledge about these risks. Many CIs stated their belief that their communities were at high risk for arsenic contamination in their groundwater, which is supported by the recent literature that showed that the highest cumulative contaminant concentrations (Arsenic, Nitrate, etc.) occurred in the San Joaquin Valley [27]. Despite expressing high general awareness about the risk for arsenic contamination in drinking water, the CIs expressed limited knowledge about the specific health risks associated with arsenic (cancer, skin lesions), similar to what was found in one 2004 study where knowledge of arsenic as a risk to health was high, but knowledge of specific health risks associated with consumption of arsenic contaminated water remained low [48]. 

Additionally, CIs expressed concerns that the outdated infrastructure of their community water system or state small water system was perpetuating poor water quality in their unincorporated rural, farming communities. Existing literature supports these concerns and shows that community water systems serving lower socioeconomic-status populations, such as the residents in Lemoore and Allensworth, are more likely to receive water contamination violations due to a lack of resources within the community to address these issues [5,20,21]. When these violations occur, those who are responsible for local water quality are required by law to let the community know. However, CIs also reported that these public violation notifications were often confusing or not understandable due to their high literate reading level and lack of clarity about the severity of reported water quality. 

One recurring barrier the CIs reported was the low literacy level of their community and the associated high literate-risk communication materials, stating that this barrier could be the reason that community residents did not know about the health risks associated with arsenic-contaminated drinking water and why they did not understand the public notices provided by the water service district. Research conducted in 2017 by the National Center for Education Statistics (NCES) for the Program for the International Assessment of Adult Competencies (ages 16–65) showed that 52% of people living in the United States had a literacy score of level 2 or below indicating intermediate or low levels of literacy [49]. According to the same study, 67% of non-native-born residents compared to 49% of native-born residents in the United States had a literacy score of level 2 or below [49]. A 2003 NCES study examined health literacy for adults and found that 88% of adults had intermediate or low levels of health literacy, and Hispanic/Latino adults had lower average health literacy across all racial/ethnic groups [50]. Health communication experts stated the importance of creating simple, direct messages to better reach diverse audiences, especially for those with low literacy levels [51,52,53]. When working with communities, health educators need to listen to the risk communication barriers expressed by community members and work with the community to identify and implement the relevant community strategies needed to raise awareness about the risk of water contaminants and sources of safe drinking water. This community-defined health promotion approach can be a pivotal point in tackling the lack of awareness about the arsenic crisis in groundwater across San Joaquin Valley. 

### 4.2. Community Suggestions for Health Education Strategies

Our results identified novel approaches to raising community awareness about water contaminant risks by providing Science, Technology, Engineering, and Math (STEM) education about drinking water safety and health to elementary school and middle school students and for the students to transfer their knowledge home. These results also identified health promotion and education approaches to community adults of color by creating linguistically inclusive materials for health communications in Spanish and indigenous languages that affirm the existing literature. The primary approach that the CIs suggested as the most effective way to inform more rural, farming community residents about the health risks associated with arsenic-contaminated drinking water, was to educate the primarily Hispanic/Latino and tribal nation farming communities’ children and adolescents first as a step toward educating their family members and other community adults. Since these STEM education courses would be designed for children, this approach would use “language brokering” between children and parents as a strategy to educate adults with limited English proficiency and/or low literacy about the risks of arsenic-contaminated drinking water. Minimal research exists examining the approach proposed by the community informants where the youth would take a STEM class about drinking water safety and health and pass knowledge along to their parents or other adults in the community. One study conducted between 2008 and 2010 in Bangladesh showed the success of a community education intervention with school children ages 8 to 11 years old [54]. This study demonstrated an increase in knowledge for students across 14 elementary schools in rural villages in Bangladesh. Among students who received the education intervention in comparison to those who did not receive the education intervention, the intervention group reported a higher rate of behavioral change in using cleaner drinking water wells [54]. Although the cited study in Bangladesh was not a community-defined education approach, we have hope that in taking the CI recommendation about educating the children as a means to educate family members and adults in the community, we could see similar results in rural, unincorporated towns in San Joaquin Valley, California when we couple a STEM education program with clean water sources.

We were not surprised that most CIs overwhelmingly stated the importance of providing community health information in both English, Spanish, and Indigenous languages to be inclusive of all community residents. The COVID-19 pandemic has made clear the importance of culturally relevant and linguistically available public health messaging to overcome linguistic barriers contributing to health disparities across communities of color where the primary language is not English, as emphasized in previous literature [32]. CIs also discussed the phenomenon of language brokering where residents who have limited English proficiency rely on others (often children or adolescent youth) to translate presentations and communication materials, including health information, from English into the minority language [55,56,57]. By developing linguistically inclusive community health information, the need for adolescent language brokering in everyday instances could be reduced. 

### 4.3. Limitations

Through the use of purposive sampling aided by our partnership with community leaders, our CIs may have been largely representative of the more involved community residents and community leaders rather than those who are less involved in the community which could have introduced selection bias. To address this type of bias, we asked all CIs if they could recommend other potential participants in line with snowball sampling. We also sought out additional CIs at a community event to recruit residents of the wider community. Since we worked closely with rural and unincorporated communities of color, our results might not be generalizable to populations in larger towns or in communities that do not reflect our study’s population demographics.

Interviews were not anonymously conducted between the interviewer and the interviewee, which may have led to social desirability bias in the responses that were collected. All responses were coded and de-identified before sharing any results outside of the research team. However, due to the nature of the interview, the researcher knew the name of the participant they were speaking with. As such, responses may have been given in a manner that seemed to be a “correct answer” instead of one that was true to the participant out of fear that answers given would be shared with community leaders or other community residents. CI interview guides were structured in such a way as to prevent leading questions that might increase the likelihood of this type of bias; however, full anonymity could not be provided during the interviews and thus social desirability bias is always a concern.

Due to the COVID-19 pandemic, all interviews were conducted over the phone or through a virtual meeting platform which might have hindered our relationship-building capacity with community informants. This also led to limitations in data collection in that we could not view body language or have interactions outside of scheduled phone calls. Additionally, the pandemic has limited our in-person interactions with our community partners which could have hindered some of the trust we were hoping to build with the community and could have skewed interview participants’ opinions about the study’s community engagement approaches.

## 5. Conclusions 

Our community assessment was designed to understand (1) the awareness and perceptions of rural, unincorporated communities of color about their water sources and the health risks of arsenic contamination from them, and (2) the barriers and preferences for risk communication and community education about drinking water safety and health. A major lesson learned for conducting research in hard-to-reach communities that we identified, and is affirmed by prior environmental health research studies [41], was that community-engaged research that prioritizes participatory action approaches, community input, and community feedback in all phases of the study are critical to identifying and developing sustainable and community relevant strategies to address clean water and other environmental health issues.

The results from this study emphasized the overwhelming need for more community-wide education about the health risks associated with arsenic contamination of drinking water. Our results provided community-defined recommendations for communication strategies including the need for health-literate, linguistically inclusive messages and developing an elementary and middle school bilingual Science, Technology, Engineering, and Math (STEM) education curriculum to increase youth, family, and community knowledge about the health risks of arsenic exposure from local drinking water. As a next step, we plan to design and translate risk communication/educational strategies identified from our results for our environmental health community education initiative about water safety and health in these communities for K-8 students, high school students, and adult community residents.

It is important to incorporate our community assessment findings into future public health intervention programs for rural, unincorporated regions that continue to rely on groundwater drinking water sources with exposure to water contaminants including, but not limited to arsenic. Additionally, it is important to take community perspectives into account for future public health interventions when there are acute water crises, such as water supply interruptions as described in a previous study [58], in addition to chronic water crises, such as the arsenic crisis in the SJV. We recommend that any public communications related to contaminated drinking water should be (1) linguistically and culturally inclusive for the predominant language and racial/ethnic communities in a specific geographic area and (2) written in easy-to-understand language for residents who have low literacy levels in English, Spanish, or another language.

For the towns we partnered with, we recommend directly implementing our study findings and beginning a language brokering approach via STEM education curriculum in the local schools to increase student, family, and community knowledge about arsenic contamination. Using a language brokering approach for risk communication that brings home the in-language information about the safety of community water systems and arsenic risk in local drinking water to the limited-English proficient and/or low-literate parents and family members from their educated children who learn about these issues of water contaminants and water safety through their STEM curriculum in the local schools. Children or students can bring home this knowledge to adults and children in their homes and improve parent, family, and community knowledge about arsenic contamination and safe drinking water. In addition, children who learn about these issues early in life will have the potential to one day transform their communities into safe water havens and sustain the knowledge and skills needed to ensure water safety in their communities into the future.

For similar rural, unincorporated towns who are at risk for drinking contaminated water, we recommend speaking with community residents and leaders to understand their perspectives and preferences for risk communication and education which may also include an educational intervention for school-aged children. Incorporating questions about residents’ primary source of drinking water (domestic wells, small community water systems, etc.), and conducting baseline knowledge testing about water contamination, might also help researchers structure appropriate intervention programs. If our future community outreach and education program approach proves effective, it could be adapted to increase awareness of additional environmental hazards among other rural, unincorporated towns, particularly those in the San Joaquin Valley of California who are at risk of drinking from contaminated water sources.

## Figures and Tables

**Figure 1 ijerph-20-00813-f001:**
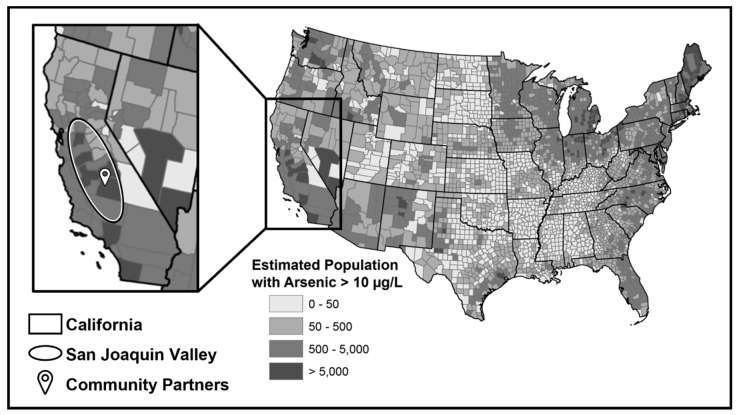
Map of populations exposed to arsenic concentration above the MCL in the United States. Inset map identifies the state of California and the study location of the San Joaquin Valley. Credit: U.S. Geological Survey Department of the Interior/USGS [28].

**Table 1 ijerph-20-00813-t001:** Community Informant Sample. N = 27.

CI Type	Job Type
Tribal Nation Leaders	Director of Education Center, Director of Recreation, Grounds Supervisor, Utilities Assistant Manager
Rural Community Leaders	Head of Community Board, Community Board Leaders, Pastor
Rural Community Residents	Farmer, Pipeline Welder, Daycare Provider, Health Clinic Worker
Local School District Members	School Board, Superintendent, Principal, Teacher, Parent
Public Water Representatives	State Water Board Member, State Water Engineer, County Environmental Health Officer, Local Water District Manager, Regional Manager, Program Manager

## Data Availability

The data presented in this study are available on request from the corresponding author. The data are not publicly available for the audio recorded interviews due to confidentiality. However, de-identified transcripts can be provided upon request.

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
