# Peer review of "Community Perceptions of Arsenic Contaminated Drinking Water and Preferences for Risk Communication in California’s San Joaquin Valley"

_ijerph, 2023, doi:10.3390/ijerph20010813_

Round 1
Reviewer 1 Report
The introduction is deficient 1) as it only focuses on water quality and water infrastructure, where no community perception investigations among similar communities somewhere else and how it benefited for intervention is lacking. 2) No information is given about the perception, and water supply information for the non-colored population is missing. The justification given for focusing on colored communities is because they are using community water utilities; what about others? Is that because the non-colored community has better knowledge or what? such disparities need to be addressed with regard to perception gaps and water quality variations, to trigger further investigations beyond this study at a global level.
Method:
It is very hard to conclude the awareness and perception of a community just with key informants. It would have been better to have quantitative data than the KI to triangulate the study. Often key informants are targeted to dig out further information about whom we think are better knowledgeable than the general community.
Author Response
Please see the attachment.
Thank you for your comments and feedback on our manuscript. On behalf of the co-authors, we want to thank you for the time and effort you put into reviewing this manuscript. Your comments have helped contribute to improving the quality of the manuscript. Please see the attachment for our responses to your comments. We want to wish you a safe and peaceful new year.

Reviewer 2 Report
The submitted manuscript concerns the important issue of community perceptions of arsenic-contaminated drinking water and preferences for risk communication in California’s San Joaquin Valley. In the research, we asked 27 key informants about their 1) available water sources, 2) knowledge about the health impacts of arsenic, and 3) preferences for risk communication and education regarding the health impacts of arsenic-contaminated drinking water. How do the authors ensure the reliability of their results? Please, justify the value of the sample. The region in which the inquiry was conducted. What's distinctive about it? What's typical, as compared to the rest of the country? What lessons should authorities draw from this analysis? Are there concrete steps that can be recommended for the authorities?
Author Response

(The authors gave the same response as above.)

Reviewer 3 Report
Thank you for your manuscript.
Given the population you studied was quite large, I feel the sample size of 27 participants which was chosen to represent this large population, does not adequately represent the population.
I think the methodology should be improved and the analyse and synthesis of the data should be done properly in a quantitative as well as a qualitative manner.
Author Response

(The authors gave the same response as above.)

Round 2
Reviewer 2 Report
The submitted manuscript concerns the important issue of community perceptions of arsenic-contaminated drinking water and preferences for risk communication in California’s San Joaquin Valley. In the research, authors asked 27 key informants about their 1) available water sources, 2) knowledge about the health impacts of arsenic, and 3) preferences for risk communication and education regarding the health impacts of arsenic-contaminated drinking water. Remarks: Thank you for your comprehensive answers. You can consider adding in the manuscript the issue about the reliability of the sample and/or credibility. In terms of the reliability of the sample, this qualitative study ensured qualitative methods standards for inter-rater reliability of the data collection and iterative data analysis process by having all the interviewers on the research team trained to recruit and conduct interviews to be able to follow the same interview protocol in an accurate way, and training these multiple research team members to conduct qualitative data analysis using the same, consistent approach. Authors also ensured each of the research team members working on the data analysis to independently conduct the data analysis and coding and then meet together to discuss and compare each other’s codes from the data analysis, and coming to agreement with the descriptive and thematic codes for the data analysis before finalizing the codebook. In addition, authors conducted substantially more than the number of informant interviews needed to reach “data saturation” or when authors have gathered as much information as authors could, with additional interviews not providing any new information or data and starting to repeat existing descriptive and thematic codes. Authors also interviewed a variety of community informants that lived or worked in these rural communities (community residents, community leaders, school district members, water service providers/experts) as part of “triangulation” of community perspectives to confirm the community issues identified were confirmed by multiple informant sources. This strengthened the credibility (a key methodological construct for qualitative research) of the results. The credibility of authors informants based on “saturation” and “triangulation” also comes from the partnership and trust authors had developed with these local communities and having these communities take initiative and ownership of the research collaboration. Add some references, and some perspective of future work, which can concern the assessment of the consumers’ perceptions of the supply of tap water in crisis situations in the section of the conclusion as presented in Consumers’ Perceptions of the Supply of Tap Water in Crisis Situations. Energies 2020, 13, 3617. The presented paper can help water managers to make informed and inclusive decisions involving a variety of factors.
Author Response
Thank you for your comments and feedback on our manuscript. On behalf of the co-authors, we want to thank you for the time and effort you put into reviewing this manuscript. Your comments have helped contribute to improving the quality of the manuscript. Please see the attachment for our responses to your comments. We want to wish you a safe and peaceful new year.

Reviewer 3 Report
Ok to be accepted.
Author Response

(The authors gave the same response as above.)
